

# Exploring *Barbronia* species diversity and phylogenetic relationship within Suborder Erpobdelliformes (Clitellata: Annelida)

Yingkui Liu[1,2], Xinxin Fu[1], Yu Wang[1], Jing Liu[1], Yong Liu[1,2], Chong Li[1,2] and Jiajia Dong[1,2]

[1] Jiangsu Key Laboratory of Brain Disease Bioinformation, Research Center for Biochemistry & Molecular Biology, Xuzhou Medical University, Xuzhou, Jiangsu, People's Republic of China
[2] School of Life Science, Xuzhou Medical University, Xuzhou, Jiangsu, People's Republic of China

Corresponding authors
Yingkui Liu,
yingkui.liu@outlook.com
Jiajia Dong,
jiajia.dong@xzhmu.edu.cn

## ABSTRACT

**Background:** *Barbronia*, a genus of freshwater macrophagous leeches, belongs to Erpobdelliformes (Salifidae: Clitellata: Annelida), and *B. weberi*, a well-known leech within this genus, has a worldwide distribution. However, the systematics of *Barbronia* have not yet been adequately investigated, primarily due to a few molecular markers, and only 20 *Barbronia* sequences available in the GenBank database. This gap significantly limits our understanding of the *Barbronia* species identification, as well as the phylogenetic placement of the genus *Barbronia* within Salifidae.

**Methods:** Next-generation sequencing (NGS) was used to simultaneously capture the entire mitochondrial genome and the full-length 18S/28S rDNA sequences. The species boundary of *Barbronia* species was estimated using bGMYC and bPTP methods, based on all available *Barbronia* COI sequences. Uncorrected COI p-distance was calculated in MEGA. A molecular data matrix consisting of four loci (COI, 12S, 18S, and 28S rDNA) for outgroups (three *Haemopis* leeches) and 49 erpobdellid leeches, representing eight genera within the Suborder Erpobdelliformes was aligned using MAFFT and LocARNA. This matrix was used to reconstruct the phylogenetic relationship of *Barbronia via* Bayesian inference (BI) and the maximum likelihood (ML) method.

**Results:** The full lengths of the mitochondrial genome, 18S and 28S rDNAs of *B.* cf. *gwalagwalensis*, are 14847 bp, 1876 bp 1876 bp, and 2863 bp, respectively. Both bGMYC and bPTP results based on COI data are generally congruent, suggesting that the previously proposed taxa (*B. arcana*, *B. weberi* formosana, and *B. wuttkei* or *Erpobdella wuttkei*) are synonyms of *B. weberi*. The specimens listed in the *B. gwalagwalensis* group, however, are split into at least two Primary Species Hypotheses (PSHs). The p-distance of the first PSH is less than 1.3% but increased to 4.5% when including the secondary PSH (*i.e.*, *B.* cf. *gwalagwalensis*). In comparison, the interspecific p-distance between the *B. weberi* group and the *B. gwalagwalensis* group ranged from 6.4% to 8.7%, and the intraspecific p-distance within the *B. weberi* group is less than 0.8%. Considering the species delimitation results and the sufficient large p-distance, the specimen sampled in China is treated as *B.* cf. *gwalagwalensis*. The monophyly of the four Erpobdelliformes families Salifidae, Orobdellidae, Gastrostomobdellidae *sensu stricto* and Erpobdellidae is well supported in ML and BI analysis based on a data of four markers. Within the Salifidae, a well-supported

*Barbronia* is closely related to a clade containing *Odontobdella* and *Mimobdella*, and these three genera are sister to a clade consisted of *Salifa* and *Linta*. According to the results of this study, the strategy of simultaneous obtaining both whole mitochondria and nuclear markers from extensively sampled Salifids species using NGS is expected to fathom both the species diversity of *B. gwalagwalensis* and the evolutionary relationship of Salifidae.

## INTRODUCTION

*Barbronia* is a genus of predatory freshwater leeches belonging to the family Salifidae (Erpobdelliformes, Clitellata, Annelida) with only three species have previously been reported in China: *Barbronia weberi*, *Barbronia zhejiangica Yang, 1996*, and *Barbronia yunnanensis Yang, Wang & Zhang, 1997* (*Yang, 1996*; *Yang, Wang & Zhang, 1997*). The former one is a well-known introduced leech species, however, the latter two are rarely known endemic *Barbronia* species. *Barbronia weberi* was originally described by *Blanchard (1897)* in Indonesia, but now it is broadly found in China (*Ta-Hsiang, 1974*; *Zhao et al., 2020*), India (*Bandyopadhyay & Mandal, 2005*; *Chandra & Mahajan, 1971*; *Ghate, 1991*), Myanmar (*Eriksen et al., 2022*), United States (*Rutter & Klemm, 2001*; *Sawyer & Sawyer, 2018*), Brazil (*Pamplin & Rocha, 2000*), Mexico (*Garduño-Montes de Oca et al., 2016*; *Oceguera-Figueroa, León-Règagnon & Siddall, 2005*), United Kingdom (*Sawyer, 1986*; *Sawyer & Sawyer, 2018*), Germany (*Kutschera, 2004*; *Nehring, 2006*), Hungary (*Ludanyi et al., 2019*), Italy (*Genoni & Fazzone, 2008*), Spain (*Pavluk, Pavluk & Rasines, 2011*), Netherlands (*Van Haaren et al., 2004*), Australia (*Govedich et al., 2003*; *Govedich, Bain & Davies, 2002*), New Zealand (*Mason, 1976*), and South Africa (*Nakano & Nguyen, 2015*). *Barbronia weberi* has achieved its current wide distribution not only through human activities (*Govedich et al., 2003*), but also through its ability to reproduce cocoons asexually, without the need for cross-fertilization (*Sawyer, 2020*). Moreover, *B. weberi* is the second intermediate host of the parasitic trematode *Australapatemon niewiadomski* (*Blasco-Costa, Poulin & Presswell, 2016*).

The taxonomic status of the widely distributed invasive species *B. weberi* needs to be evaluated on a global scale (*Oceguera-Figueroa et al., 2011*), which involved *B. weberi formosana*, *B. wuttkei*, *B. arcana*, and *Barbronia* sp. Molecular identification has become a valuable complement to morphological taxonomy over the past two decades. A partial mitochondrial gene COI sequence has been used to recognize *Barbronia* species. *Barbronia wuttkei* was originally described as *Erpobdella wuttkei* based solely on morphological traits (*Kutschera, 2004*), however, it was later transferred to the genus *Barbronia* but retained as a valid name by *Grosser & Trontelj (2008)* based on COI phylogenetic evidence. The genetic analysis using the COI sequence-based species delimitation approach, Poisson Tree Process (PTP), has led to the initial assumption that *Barbronia wuttkei* and *B. arcana*

should be classified under the taxon *B. weberi* (*Klass et al., 2021*). One *Barbronia* sp. (GenBank accession: MN503261), as well as another specimen (GenBank accession: MF458701, marked as *Erpobdella* sp.), are recently recognized as *B. gwalagwalensis* on basis of COI data (*Klass et al., 2021*).

The phylogeny of *Barbronia* was initially inferred solely based on 18S ribosomal DNA sequences alone, *B. weberi* was found to be closely related to a clade of two erpobdellids: *Dina lineata* (synonym: *Erpobdella lineata*) and *Erpobdella octoculata* (*Trontelj, Sket & Steinbrück, 1999*). Although an attempt was made to amplify a fragment of the mitochondrial marker 12S rDNA from the *B. weberi* specimen by *Trontelj, Sket & Steinbrück (1999)*, it failed using the primer pair 12S-A/12S-B (*Siddall, 2002*). Consequently, only the unique 18S rDNA sequence of *B. weberi* was used in subsequent phylogenetic study (*Borda & Siddall, 2004a*; *Oceguera-Figueroa, León-Règagnon & Siddall, 2005*). A comprehensive phylogenetic analysis was conducted using combined data from partial sequences of 18S rDNA, 28S rDNA, 12S rDNA, and cytochrome c oxidase subunit I (COI) published by *Borda & Siddall (2004b)*. This study confirmed that the *Barbronia* species (*B. weberi*, *B. weberi formosana*, *B. gwalagwalensis*, and an undescribed *Barbronia* species) formed a monophyletic group closely related to *Linta be*. Another phylogenetic result supported that *B. weberi* and *B. arcana* were part of the Salifidae clade, a sister group of Erpobdellidae (*Oceguera-Figueroa, León-Règagnon & Siddall, 2005*). An expanded dataset including six *Barbronia* specimens represented all mentioned *Barbronia* species and *B. wuttkei*, forming a monophyletic group that was a sister group to a clade of *Salifa* and *Linta* species (*Oceguera-Figueroa et al., 2011*). Recently, *Barbronia* represented by three taxa (*Anderson, Braoudakis & Kvist, 2020*), and *Barbronia weberi* only (*Tessler, Siddall & Oceguera-Figueroa, 2018*), are closely related to a clade that includes *Odontobdella blancharidi* and *Mimobdella japonica*.

The implementation of next-generation sequencing (NGS) technologies has led to a substantial increase in the number of whole mitochondrial genomes in the GenBank database (*Kuang & Yu, 2019*), and has gradually become a mainstay of species identification and phylogeny research (*Franco-Sierra & Diaz-Nieto, 2020*; *Kortsinoglou et al., 2020*). To date, however, only about 20 sequences of various gene fragments from *Barbronia* specimens are available in the GenBank database. Eight of these sequences are the mitochondrial COI barcodes, six partial 18S rDNA sequences, four short 28S rDNA sequences, and only two 12S rDNA sequences, representatively. Neither a complete mitochondrial genome of *Barbronia* species nor even the family Salifidae has been published. This gap significantly limits our understanding of either the species identification or phylogenetic relationships of *Barbronia* within Salifidae.

In the present study, the whole mitochondrial genome, as well as the 18S rDNA and 28S rDNA of *Barbronia* cf. *gwalagwalensis*, are assembled and annotated; the species boundary of *Barbronia* species is evaluated using two kinds of single-locus COI based species delimitation methods; the phylogenetic relationships of *Barbronia* within the Suborder Erpobdelliformes are conducted based on both mitochondrial and nuclear data.

## MATERIALS AND METHODS

### DNA extraction and assembly of the mitochondrial genome, and rDNAs

The specimen later identified as *Barbronia* cf. *gwalagwalensis* was sampled using a hydrobiological Peterson dredge during a general benthic sampling expedition in Yunlong Lake (Xuzhou, Jiangsu province, China), and morphological identification was performed with reference to *Yang (1996)*. The total DNA material was extracted using Ezup Column Animal Genomic DNA Purification Kit (Sangon Biotech Company, Shanghai, China) following the manufacturer's protocols. The quality of DNA was evaluated *via* gel electrophoresis and visualized on a ChemiDoc XRS + (Bio-Rad, Hercules, CA, USA). The quantification of DNA was checked *via* the Spectrophotometer NanoDrop 2000. DNA was eluted in ddH$_2$O and stored at $-20$ °C until used for next-generation sequencing (NGS). A sequencing library was constructed using a Nextera XT DNA Library Preparation Kit (Illumina, San Diego, CA, USA), and sequencing reads were generated on a MiSeq System (Illumina, San Diego, CA, USA) to generate 150 bp paired-end reads. Low-quality bases in raw FastQ reads were trimmed using Trimmomatic v.0.36 with default setting, the remaining reads were assembled as contig and used to reconstruct the mitochondrial genome *via* GetOrganelle V1.1.7 with recommended setting (*Jin et al., 2020*). The annotation of mitochondrial protein-coding genes, Transfer RNAs, and ribosomal DNAs of *B.* cf. *gwalagwalensis* was performed by selecting the Invertebrate genetic code and referencing the RefSeq63 Metazoa dataset *via* MITOS2 Webserver. Additionally, manual checks were conducted by comparing nucleotide sequences with those from the published available closely related leech species to refine these annotations when possible using Geneious 2021.

### Single-locus based species delimitation

Species delimitation of *Barbronia* species should be assessed using various molecular species delimitation approaches with data from single-locus or multiple loci. Only 20 *Barbronia* sequences are available in GenBank, and half of these sequences are partial mitochondrial gene COI sequences (Table 1). In addition, two specimens (accession number: DQ009666 and MF458701) labeled as *Erpobdella* sp. are also included, because both are used in previous studies of *Barbronia* (*Anderson, Braoudakis & Kvist, 2020*; *Grosser & Trontelj, 2008*; *Klass et al., 2021*; *Oceguera-Figueroa et al., 2011*). The species delimitation of *Barbronia* was conducted solely based on a single-locus dataset of COI, using both a Bayesian implementation of the general mixed yule-coalescent model (bGMYC) and an updated version of the maximum likelihood Poisson Tree Processes model (bPTP) in the current study. The bGMYC model is capable of inferring the transition between the Yule model (species level) and coalescent model (population level within species) (*Fujisawa & Barraclough, 2013*), with posterior probabilities accounting for phylogenetic uncertainty (*Reid & Carstens, 2012*). The bPTP analysis required a phylogenetic gene tree, and putative species were estimated with Bayesian support through a simulation of Poisson Tree Processes (*Zhang et al., 2013*).

**Table 1 Listing all COI sequences used for species delimitation and other published *Barbronia* data in GenBank.**

| Groups | Taxon | Country | COI accession | COI size (bp) | 12S accession | 18S accession | 28S accession |
|---|---|---|---|---|---|---|---|
| *Barbronia weberi* | *B. weberi* | Costa Rica | HQ336339 | 649 | – | AF099951 | HQ336356 |
| | *B. weberi* formosana | United States | AY786456 | 600 | – | AY786461 | AY786448 |
| | "*B. arcana*" | Mexico | DQ235598 | 649 | DQ235588 | DQ235608 | – |
| | *B. weberi* | Mexico | KU553102 | 645 | – | – | – |
| | *B. weberi* | Mexico | KU553103 | 645 | – | – | – |
| | "*Erpobdella wuttkei*" | Germany | DQ009666 | 584 | – | – | – |
| | *Barbronia* sp | South Africa | AY786457 | 360 | – | AY786463 | AY786450 |
| *Barbronia gwalagwalensis* | *B.* cf. *gwalagwalensis* | China | **OQ339201**\* | 1,534 | **OQ339201**\* | **OQ269483**\* | **OQ269482**\* |
| | *B. gwalagwalensis* | South Africa | AY786455 | 480 | – | AY786462 | AY786449 |
| | *Barbronia* sp | Myanmar | MN295405 | 660 | – | – | – |
| | *Barbronia* sp | South Korea | MN503261 | 710 | – | MT010330 | – |
| | "*Erpobdella*. sp" | France | MF458701 | 634 | – | – | – |
| Outgroups | *Mimobedlla japonica* | Japan | AB679658 | 1,267 | – | – | – |
| | *Odontobdella blancharidi* | Japan | AB938004 | 1,267 | – | – | – |

Note:
The groups referred to the clade A, clade B and outgroups of the COI tree in Fig. 3, respectively. Taxon names enclosed in quotations indicate that these taxa may have been misidentified. The accession numbers highlighted with asterisks in bold indicate newly added data., and a dash ("−") indicates that no sequences are available.

For species delimitation analyses in bGMYC and bPTP, the input COI phylogenetic tree was performed with BEAST Version 1.10.4, using HKY + I + G substitution model with two partitions (one partition comprised the 1st and 2nd codon positions of COI, the other one was the 3rd codon position) with a yule speciation prior and a strict clock. A total of ten million MCMC generations were performed, with samples taken every 1,000 generations. After burn-in, a maximum clade credibility tree was built using TreeAnnotator. The bGMYC analyses were conducted following the approach proposed by *Liu et al. (2017)*. The bGMYC analyses consisted of the total 50,000 MCMC generations, with a thinning interval of every 100 generations, discarding the first 2,000 generations as burn-in, and setting the upper and lower bounds on the threshold parameter from 1 (the minimum number of species) to 11 (the maximum number of tips in COI tree). For bPTP analyses were performed with default setting *via* an online server (https://species.h-its.org/). In addition, it is worth noting that GMYC models may not provide accurate delineate species properly in data sets composed of only one or two species (*Dellicour & Flot, 2015*), therefore, *Mimobedlla japonica* and *Odontobdella blancharidi* were included in the analyses of species delimitation.

## Phylogenetic analysis using multiple loci

To corroborate previous overarching phylogenetic frameworks, 261 sequences of four molecular markers (COI, 12S rDNA, 18S rDNA, and 28S rDNA) representing 49 erpobdellid leeches and three *Haemopis* leeches were retrieved from the GenBank database (Table 2). The genera *Dina* and *Mooreobdella* are synonyms of the genus *Erpobdella* (*Siddall, 2002*), consequently, both *Dina* and *Mooreobdella* are referred as *Erpobdella* in the
**Table 2 A list of erpobdellids leech specimens and GenBank accession number of four molecular marks used in the current phylogeny reconstruction.**

| Family | Species | COI | 12S | 18S | 28S |
|---|---|---|---|---|---|
| Salifidae | *Barbronia* cf. *gwalagwalensis* | **OQ339201*** | **OQ339201*** | **OQ269483*** | **OQ269482*** |
| Salifidae | *Barbronia gwalagwalensis* | AY786455 | – | AY786462 | – |
| Salifidae | *Barbronia weberi* | HQ336339 | DQ235588 | AF099951 | HQ336356 |
| Salifidae | *Odontobdella blanchardi* | AB675016 | AB675017 | AB663651 | AB663671 |
| Salifidae | *Mimobdella japonica* | AB675014 | AB675015 | AB663650 | AB663670 |
| Salifidae | *Linta be* | AY786460 | – | AY786466 | AY786453 |
| Salifidae | *Salifa motokawai* | LC029431 | LC029432 | LC029434 | LC274548 |
| Salifidae | *Salifa perspicax* | HQ336341 | HQ336349 | HQ336375 | HQ336359 |
| Salifidae | *Salifa perspicax* | HQ336343 | HQ336351 | HQ336377 | HQ336360 |
| Orobdellidae | *Orobdella angustata* | LC323139 | LC323141 | LC323140 | LC431606 |
| Orobdellidae | *Orobdella brachyepididymis* | LC106320 | LC106318 | LC106319 | LC274535 |
| Orobdellidae | *Orobdella dolichopharynx* | AB675028 | AB675029 | AB663665 | AB663666 |
| Orobdellidae | *Orobdella esulcata* | AB675020 | AB675021 | AB663655 | AB663656 |
| Orobdellidae | *Orobdella ghilarovi* | LC431609 | LC431616 | LC431608 | LC431607 |
| Orobdellidae | *Orobdella ijimai* | AB675030 | AB675031 | AB663659 | AB663660 |
| Orobdellidae | *Orobdella kanaekoikeae* | LC184548 | LC184547 | LC184551 | LC274533 |
| Orobdellidae | *Orobdella kawakatsuorum* | AB675032 | AB675033 | AB663661 | AB663662 |
| Orobdellidae | *Orobdella ketagalan* | AB704787 | AB704788 | AB704785 | LC274546 |
| Orobdellidae | *Orobdella koikei* | AB679688 | AB679689 | AB698883 | LC274543 |
| Orobdellidae | *Orobdella masaakikuroiwai* | AB938006 | AB937997 | AB938003 | LC274530 |
| Orobdellidae | *Orobdella meisai* | LC314424 | LC314422 | LC314423 | LC431605 |
| Orobdellidae | *Orobdella mononoke* | AB698866 | AB698867 | AB698868 | LC274547 |
| Orobdellidae | *Orobdella montipumila* | LC616663 | LC616667 | LC616674 | LC616673 |
| Orobdellidae | *Orobdella nakahamai* | LC106331 | LC106329 | LC106330 | LC274534 |
| Orobdellidae | *Orobdella naraharaetmagarum* | LC087144 | LC087142 | LC087143 | LC274531 |
| Orobdellidae | *Orobdella octonaria* | AB675024 | AB675025 | AB663667 | AB663668 |
| Orobdellidae | *Orobdella octonaria* | HQ336338 | HQ336348 | HQ336372 | HQ336355 |
| Orobdellidae | *Orobdella okanoi* | LC106342 | LC106340 | LC106341 | LC274532 |
| Orobdellidae | *Orobdella shimadae* | AB675026 | AB675027 | AB663663 | AB663664 |
| Orobdellidae | *Orobdella tsushimensis* | AB675018 | AB675019 | AB663653 | AB663654 |
| Orobdellidae | *Orobdella whitmani* | AB675022 | AB675023 | AB663657 | AB663658 |
| Orobdellidae | *Orobdella yamaneae* | LC106350 | LC106348 | LC106349 | LC274536 |
| Gastrostomobdellidae *sensu stricto* | *Gastrostomobdella ampunganensis* | LC274551 | LC274564 | LC274517 | LC274516 |
| Gastrostomobdellidae *sensu stricto* | *Gastrostomobdella ampunganensis* | LC274559 | LC274568 | LC274525 | LC274524 |
| Gastrostomobdellidae *sensu stricto* | *Gastrostomobdella extenta* | LC274553 | LC274565 | LC274519 | LC274518 |
| Gastrostomobdellidae *sensu stricto* | *Gastrostomobdella extenta* | LC274555 | LC274566 | LC274521 | LC274520 |
| Gastrostomobdellidae *sensu stricto* | *Gastrostomobdella extenta* | LC274557 | LC274567 | LC274523 | LC274522 |
| Gastrostomobdellidae *sensu stricto* | *Gastrostomobdella monticola* | AB675011 | AB675010 | AB663649 | AB663669 |
| Gastrostomobdellidae *sensu stricto* | *Gastrostomobdella monticola* | LC274549 | LC274563 | LC274514 | LC274513 |
| Erpobdellidae | *Dina lineata* | – | AF099952 | AF099950 | – |
| Erpobdellidae | *Erpobdella adani* | MG745144 | MG745141 | MG745138 | – |

| Table 2 (continued) | | | | | |
|---|---|---|---|---|---|
| Family | Species | COI | 12S | 18S | 28S |
| Erpobdellidae | *Erpobdella adani* | MG745145 | MG745142 | MG745139 | – |
| Erpobdellidae | *Erpobdella adani* | MG745146 | MG745143 | MG745140 | – |
| Erpobdellidae | *Erpobdella annulata* | HQ336345 | – | HQ336379 | HQ336362 |
| Erpobdellidae | *Erpobdella bucera* | MN612829 | MN613043 | MN613063 | MN613084 |
| Erpobdellidae | *Erpobdella costata* | AY425460 | AY425442 | AY425478 | AY425406 |
| Erpobdellidae | *Erpobdella dubia* | AF116023 | AF462022 | AF115997 | AY425365 |
| Erpobdellidae | *Erpobdella japonica* | AB675012 | AB675013 | AB663648 | AB663652 |
| Erpobdellidae | *Erpobdella japonica* | AF116026 | AF462023 | AF116000 | AY425366 |
| Erpobdellidae | *Erpobdella mexicana* | DQ235595 | DQ235585 | DQ235605 | HQ336364 |
| Erpobdellidae | *Erpobdella mexicana* | DQ235597 | DQ235587 | DQ235607 | HQ336365 |
| Erpobdellidae | *Erpobdella microstoma* | MN612934 | MN613044 | MN613065 | MN613086 |
| Erpobdellidae | *Erpobdella montezuma* | GQ368760 | GQ368820 | GQ368802 | – |
| Erpobdellidae | *Erpobdella obscura* | MN613005 | MN613045 | MN613066 | MN613087 |
| Erpobdellidae | *Erpobdella obscura* | MN612911 | MN613046 | MN613067 | MN613088 |
| Erpobdellidae | *Erpobdella ochoterenai* | DQ235596 | DQ235586 | DQ235606 | HQ336370 |
| Erpobdellidae | *Erpobdella ochoterenai* | DQ235599 | DQ235590 | DQ235609 | HQ336371 |
| Erpobdellidae | *Erpobdella octoculata* | HQ336344 | – | HQ336378 | HQ336361 |
| Erpobdellidae | *Erpobdella parva* | MN612997 | MN613052 | MN613073 | MN613094 |
| Erpobdellidae | *Erpobdella parva* | MN612930 | MN613053 | MN613074 | MN613095 |
| Erpobdellidae | *Erpobdella punctata* | HQ336346 | HQ336352 | HQ336380 | HQ336363 |
| Erpobdellidae | *Erpobdella punctata* | MN612994 | MN613056 | MN613077 | MN613098 |
| Erpobdellidae | *Erpobdella testacea* | AF116027 | AF462025 | AF116003 | AY425370 |
| Erpobdellidae | *Erpobdella triannulata* | DQ235602 | DQ235592 | DQ235612 | HQ336366 |
| Erpobdellidae | *Erpobdella triannulata* | HQ336347 | HQ336353 | DQ235614 | HQ336367 |
| Erpobdellidae | *Mooreobdella melanostoma* | AF116025 | AF462027 | AF115999 | AY425395 |
| Haemopidae | *Haemopis caeca* | AY040702 | AY425419 | AY040687 | AY425376 |
| Haemopidae | *Haemopis kingi* | OK447797 | OK489304 | OK489319 | OK486532 |
| Haemopidae | *Haemopis marmorata* | OK447840 | OK489305 | OK489313 | OK486525 |

**Note:**
The accession numbers highlighted with asterisks in bold indicate newly added data, and a dash ("–") indicates that no sequences are available.

current study. COI sequences were aligned using MAFFT 7.45 with the most accurate LINSI option (*Katoh & Standley, 2014*), and sequences of rDNA were aligned *via* LocARNA with default settings (*Will et al., 2012*). The individual alignments of four markers were concatenated in PhyloSuite (*Zhang et al., 2020*) as a matrix used for flowing analyses. Phylogeny analyses were performed separately using Bayesian inference (BI) and the maximum likelihood (ML) method *via* MrBayes V3.2.7 (*Altekar et al., 2004*) and IQ-TREE (*Nguyen et al., 2015*). *Haemopis caeca*, *Haemopis kingi*, and *Haemopis marmorata* belonging to Haemopidae (Hirudiniformes) were used as outgroups. The best-fit evolutionary model of each molecular marker (COI, 12S rDNA, 18S rDNA and 28S rDNA) data was determined using PartitionFinder 2 with the 'greedy' algorithm

based on the AICc score (*Lanfear et al., 2017*). For Bayesian analysis, two independent runs, with four Markov Chain Monte Carlo (MCMC) chains each, were simultaneously carried out for four million generations and sampled every 10,000 generations. The analysis was assumed to have reached stationarity when the potential scale reduction factor value (PSRF) approached 1.0 and the effective sample size value >100. After discarding the 25% samples as burn-in, the 50% majority-rule consensus tree was built. For the ML analysis, the reliability of bootstrap values and tree topology was assessed by ultrafast bootstrap using 1,000 replicates.

## RESULTS

### General features of mitochondrial genomes and nuclear rDNAs

The newly assembled mitochondrial genome of *B.* cf. *gwalagwalensis* is AT-rich (71.9%) circular mapping molecule (total length, 14,847 bp), with an average coverage of 209-fold. The components of *B.* cf. *gwalagwalensis* mitogenome consist of 13 protein-coding genes (PCGs), 22 transfer RNAs (tRNAs), 2 rDNAs (12S and 16S rDNA), and a possible control region (451 bp, the longest non-coding reign is located between tRNA R and tRNA H). All mitochondrial genes of *Barbronia* are encoded on the same strand, and gene order are generally consistent with previously published erpobdellid mitogenomes (Fig. 1 and Table 3). Among 13 *Barbronia* PCGs, both NAD2 and NAD5 genes are inferred to use ATT as an initiation codon, the COX3 gene is initiated with ATA, and the remaining genes used ATG as a start codon. The predicted secondary structures of 22 tRNAs had a similar clover leaf shape (Fig. 2), and these tRNAs range in size from 60 to 69 bp, with the shortest tRNA gene being tRNA L and the longest one being tRNA Q. The complete length of nuclear 18S rDNA and 28S rDNAs are 1,876 and 2,863 bp, respectively, the former one is identical but longer than the partial 18S rDNA sequence (AY786462) of *B. gwalagwalensis*.

### The outcome of single-locus-based species delimitation

The *B. weberi* groups and the *B. gwalagwalensis* group listed in Table 1 correspond to two main well-supported (PP > 0.99) clades in COI phylogenetic analysis (Fig. 3). All members of the *B. weberi* group are consistently recognized as a well-supported PSH, including the previous described taxa *B. weberi*, *B. weberi* formosana, *B. arcana*, *Barbronia wuttkei* (clade A in Fig. 3). The primary species hypothesis (PSH) of *B. weberi* groups proposed by bPTP are consistent with bGMYC, but the difference between two results is the species delimitation of individuals listed in the *B. gwalagwalensis* group (clade B in Fig. 3). In the bPTP results, the five individuals of the *B. gwalagwalensis* group in clade B are divided into two PSHs. One well supported PSH comprise only *B.* cf. *gwalagwalensis*, and the other moderately supported PSH encompass the paratype of *B. gwalagwalensis* (GenBank accession number: AY786455) and the remaining specimens (MF458701, MN295405 and MN503261, Fig. 3 and Table 1). However, the five individuals of the *B. gwalagwalensis* group are either split into four well-supported PSHs or collectively classified as one moderate PSH in the bGMYC analysis (Fig. 3). In the four well-supported PSHs, one PSH contains MF458701 and AY786455, and each of the remaining PSHs is formed separately from MN295405, MN503261and OQ339201. *Barbronia* cf. *gwalagwalensis* (OQ339201)

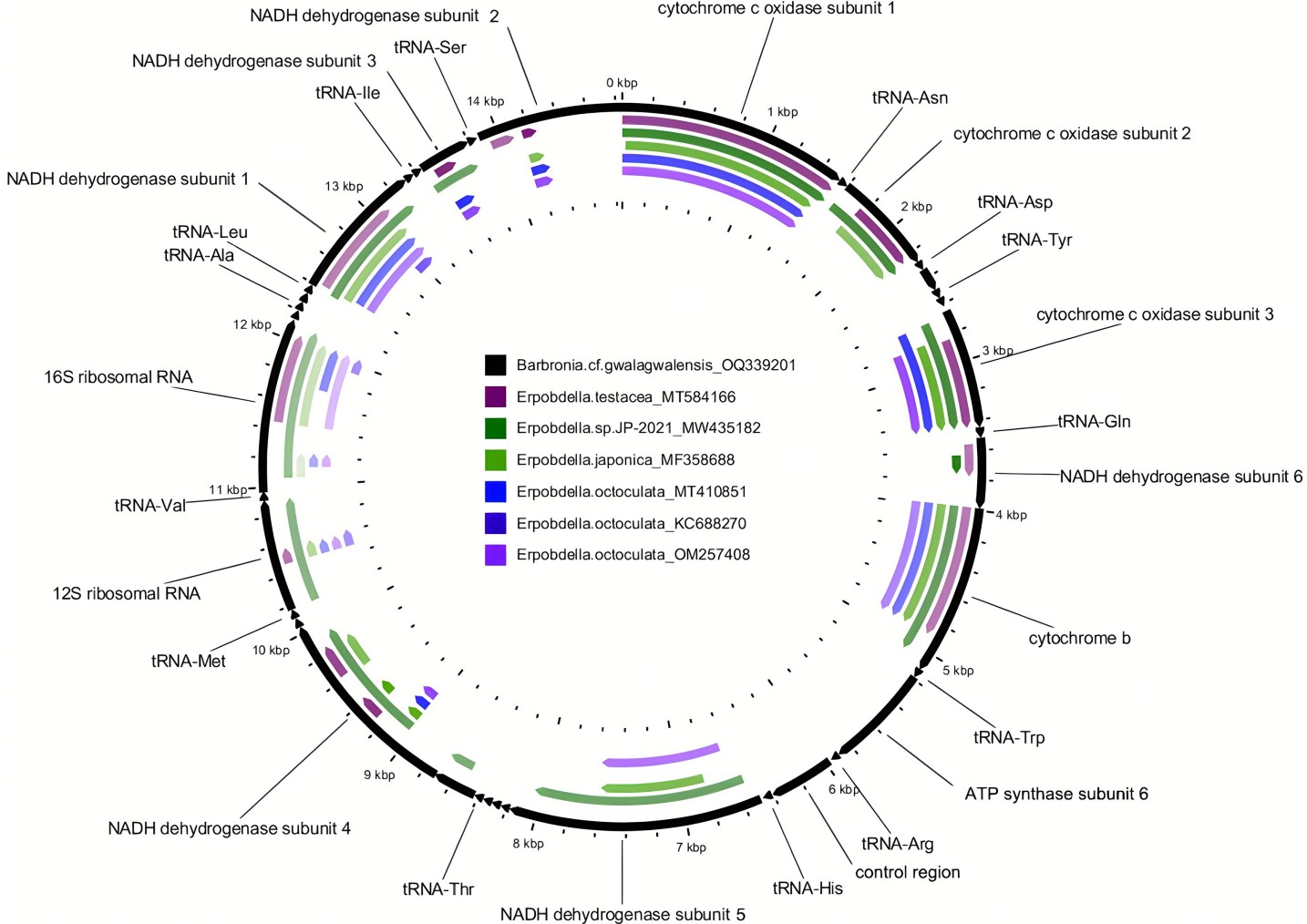

**Figure 1 Comparison of the newly assembled mitochondrial genome (OQ339201) of *Barbronia* cf. *gwalagwalensis* with six previously published erpobdellids using BLASTn.** The outermost slot (in black) displays 22 tRNAs, the 12S and 16S rDNA, and 13 coding gene regions on the mitochondrial genome (14,847 base pairs in length). Each of the six inner slots represents regions of the mitochondrial genome (OQ339201) where a BLAST hit occurred in one of the six erpobdellid specimens. Empty regions indicate where there were no BLAST hits between the newly assembled mitochondrial genome and the six previously published erpobdellid specimens using an expect value cutoff of 1e-10, alignment length cutoff of 100, and percent identity cutoff of 80%.

**Table 3 General features of the mitochondrial genomes of *B.* cf. *gwalagwalensis* and other specimens.**

| Taxa | Size (bp) | %GC | CDSs | tRNAs | rRNAs | Accession |
|------|-----------|-----|------|-------|-------|-----------|
| ***B.* cf. *gwalagwalensis*** | 14,847 | 28.1 | 13 | 22 | 2 | OQ339201* |
| *Erpobdella testacea* | 14,495 | 26.9 | 13 | 20 | 2 | MT584166 |
| *Erpobdella japonica* | 14,725 | 27.9 | 13 | 22 | 2 | MF358688 |
| *Erpobdella octoculata* | 13,035 | 27.9 | 12 | 20 | 2 | MT410851 |

(Continued)

| Table 3 | (continued) | | | | | |
|---|---|---|---|---|---|---|
| Taxa | Size (bp) | %GC | CDSs | tRNAs | rRNAs | Accession |
| *Erpobdella octoculata* | 15,580 | 27.5 | 13 | 22 | 2 | OM257408 |
| Erpobdellidae sp. | 14,746 | 30.2 | 13 | 22 | 2 | MW43152 |
| "*Erpobdella octoculata*" | 14,407 | 28.4 | 13 | 22 | 2 | KC688270 |

**Note:**
The specimen (KC688270 or NC_023927, named as "*Erpobdella octoculata*") is likely to be a misidentified taxon or the outcome of sequencing contamination. The accession numbers highlighted with asterisks in bold indicate newly added data.

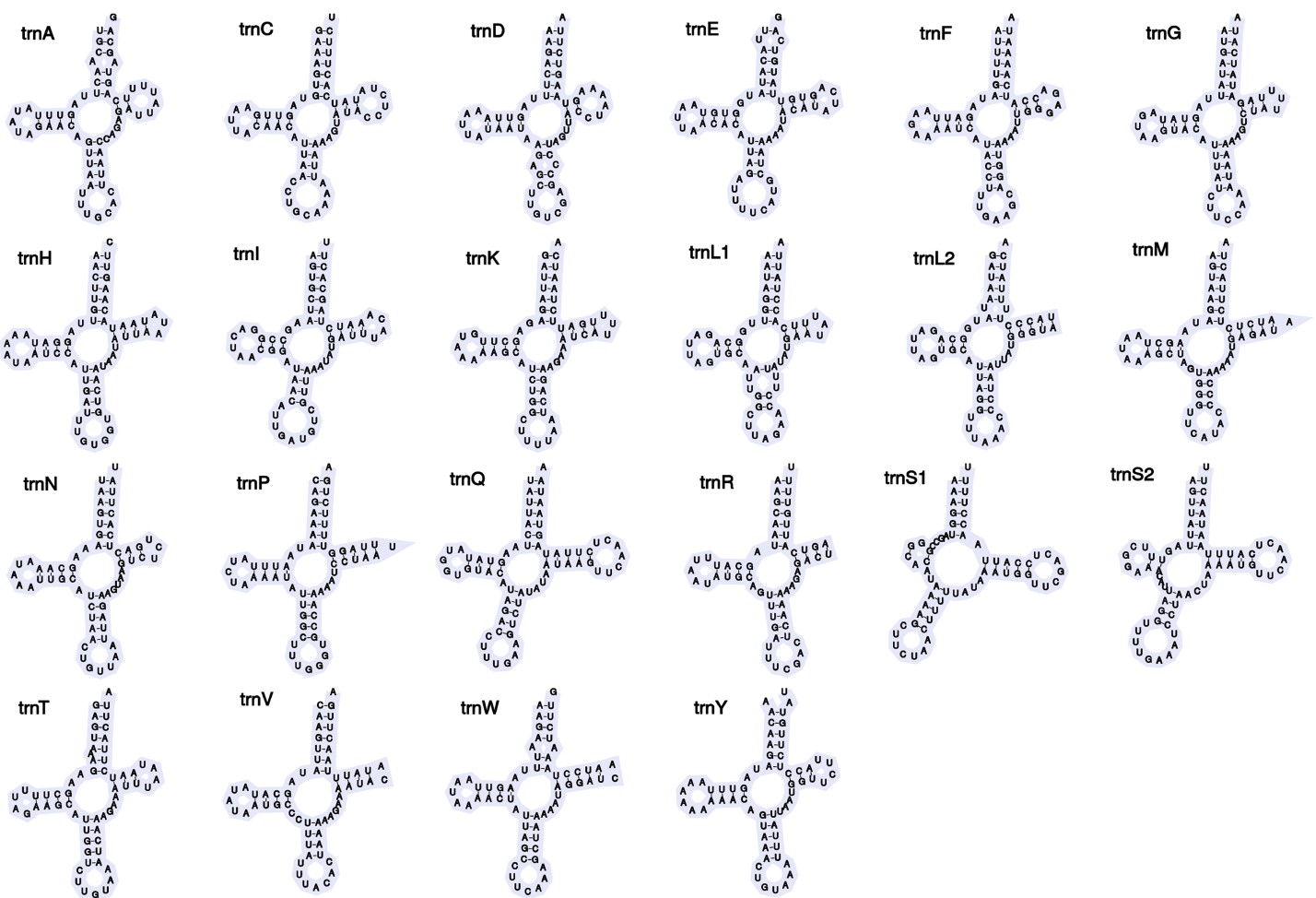

**Figure 2** Secondary structures of tRNA genes in the mitogenome of *Barbronia* cf. *gwalagwalensis*.

alone is consistently recognized as a separately well-supported PSH in both bGMYC and bPTP analyses (threshold 0.95–1), although it is also somewhat classified with the paratype *B. gwalagwalensis* (AY786455) in a moderately supported PSH (threshold 0.5–0.9) by bGMYC. Moreover, the uncorrected p-distance of COI sequences between *B.* cf. *gwalagwalensis* (OQ339201) and the paratype *B. gwalagwalensis* (AY786455) is 4.4% (Fig. 4), which is substantially higher compared to the p-distance (ranged from 0.2% to 1.2%) among four individuals in clade B (Fig. 4) collected from Myanmar (MN295405),

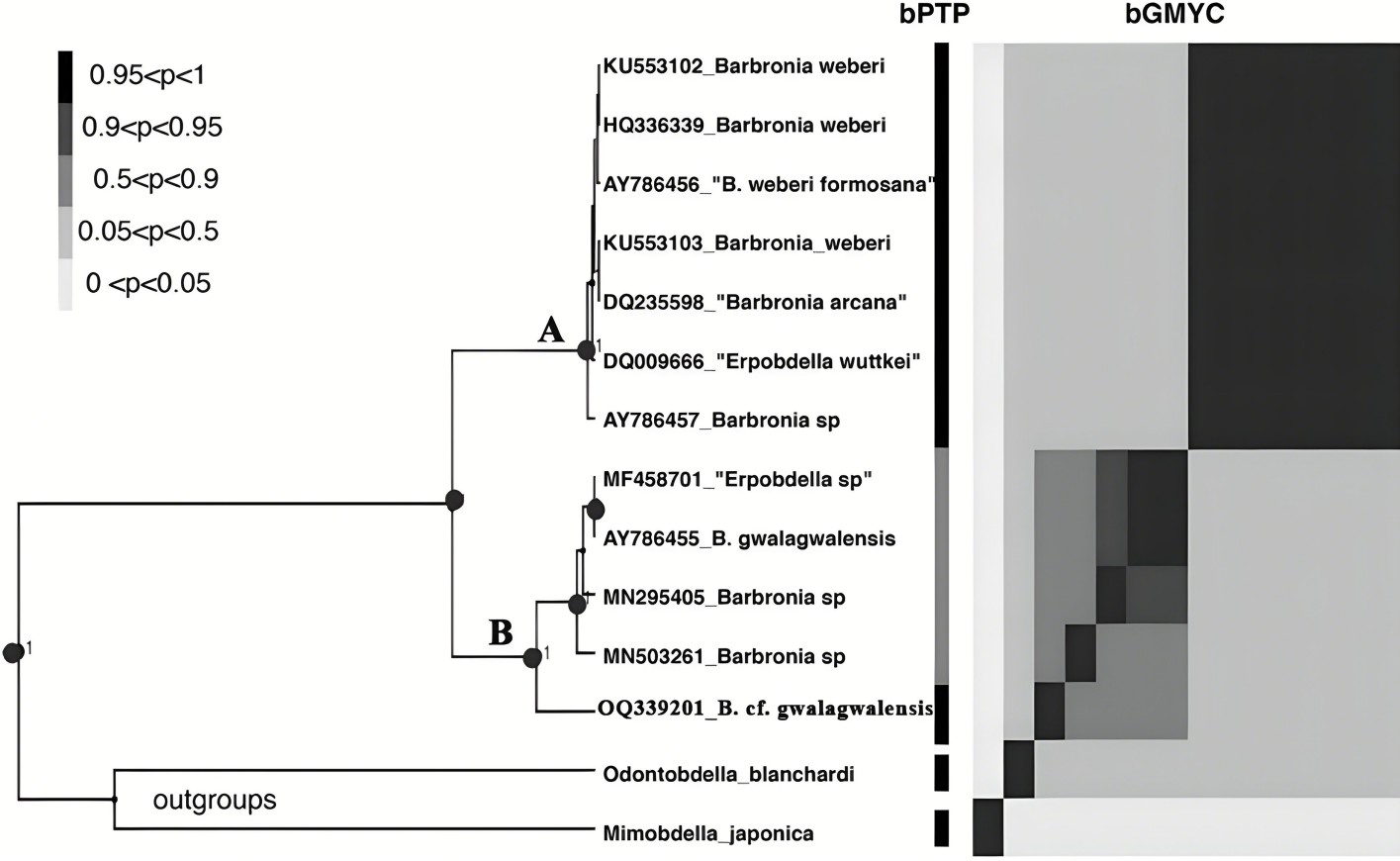

**Figure 3 The results of the primary species hypothesis (PSHs) based on COI data.** The bPTP and bGMYC results were summarized and visualized, the left side was the maximum clade credibility tree from BEAST analyses, and the species delimitation results, using bPTP and bGMYC methods based on COI data, were showed on the right side, with colors corresponding to the posterior probability of same primary species hypotheses (PSHs) under a specific threshold (at the upper left). The accession numbers and taxon names in GenBank presented besides underlines at the tip of the tree. Clade A and clade B referred to the *B. weberi* group and the *B. gwalagwalensis* group, respectively.

South Korea (MN503261), South Africa (AY786455) and France (MF458701) (Fig. 4 and Table 1). The p-distance between above four individuals and *B. cf. gwalagwalensis* ranged from 3.8% to 4.5%. In contrast, the p-distances of seven *B. weberi* individuals in clade A (Fig. 3) sampled from South Africa, the United States, Mexico, Costa Rica, and Germany are less than 0.8%, and the interspecific uncorrected p-distance between the *B. weberi* group and the *B. gwalagwalensis* group (clade A and clade B, respectively, Fig. 3) ranged from 6.4% to 8.7%. Considering the result of species delimitation and the sufficiently large p-distance mention above, the current specimen (OQ339201) is treated as *B. cf. gwalagwalensis*.

## Phylogenetic relationships within Erpobdelliformes

The best-fit model for each partition, GTR + I + G, was selected using PartitionFinder 2. Three *Haemopis* leeches belonging to Haemopidae (Hirudiniformes) were used as outgroups both in Bayesian Inference (BI) analysis and the maximum likelihood (ML) analysis. Both the BI tree and ML tree topologies show generally well-supported for four

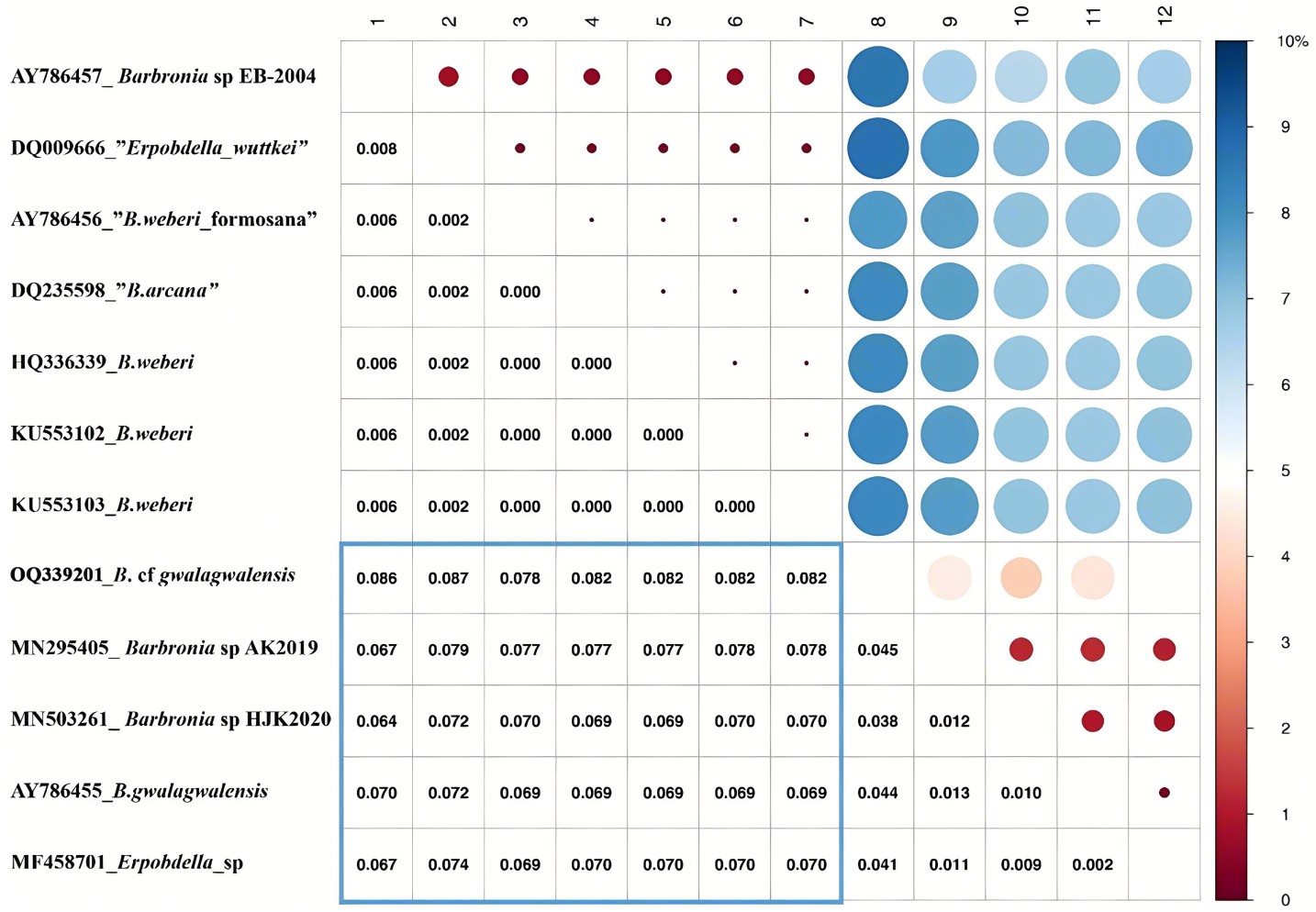

**Figure 4 Summary of the bGMYC and bPTP species delimitation methods using the mitochondrial data set.** The uncorrected p-distance matrix corresponding to the alignment of the Barbronia COI sequences. The uncorrected p-distance was calculated in MEGA X using the pairwise deletion option, the uncorrected p-distances were visualized in the upper triangular portion of this matrix, with a color bar (0~10% uncorrected p-distance). The intraspecific and interspecific uncorrected p-distances were represented by the red and blue circles, and the size of circles indicate the value of the corresponding uncorrected p-distances which were listed in triangular portion of this matrix. At the left, the accession numbers and taxon names in GenBank presented besides underlines, corresponding to the numbers from 1 to 12 at the top.

main monophyletic groups (PP = 1.00, and BS > 95%), despite differences in the placement of the four clades (Fig. 5). The first clade (PP = 1.00, and BS = 99%) is Salifidae represented by nine specimens from five genera (*Barbronia*, *Mimobdella*, *Odontobdella*, *Linta*, and *Salifa*), the second clade (PP = 1.00, and BS = 99%) is Orobdellidae consisted of 18 *Orobdella* species, the third one (PP = 1.00, and BS = 95%) is Gastrostomobdellidae *sensu stricto* on behalf of three *Gastrostomobdella* species, and the last one is a well-supported Erpobdellidae clade (PP = 1.00, and BS = 100%) of 18 *Erpobdella* species. *Barbronia* is a well-supported (PP = 1.00, BS = 100%) monophyletic group, and its closely related clade consisted of *Mimobdella* and *Odontobdella*. The genus *Salifa* is recovered as paraphyletic with the inclusion of *Linta be*. Gastrostomobdellidae *sensu stricto* (containing genus *Orobdella* only), but not Gastrostomobdellidae *sensu lato* (*Orobdella* + *Gastrostomobdella*),

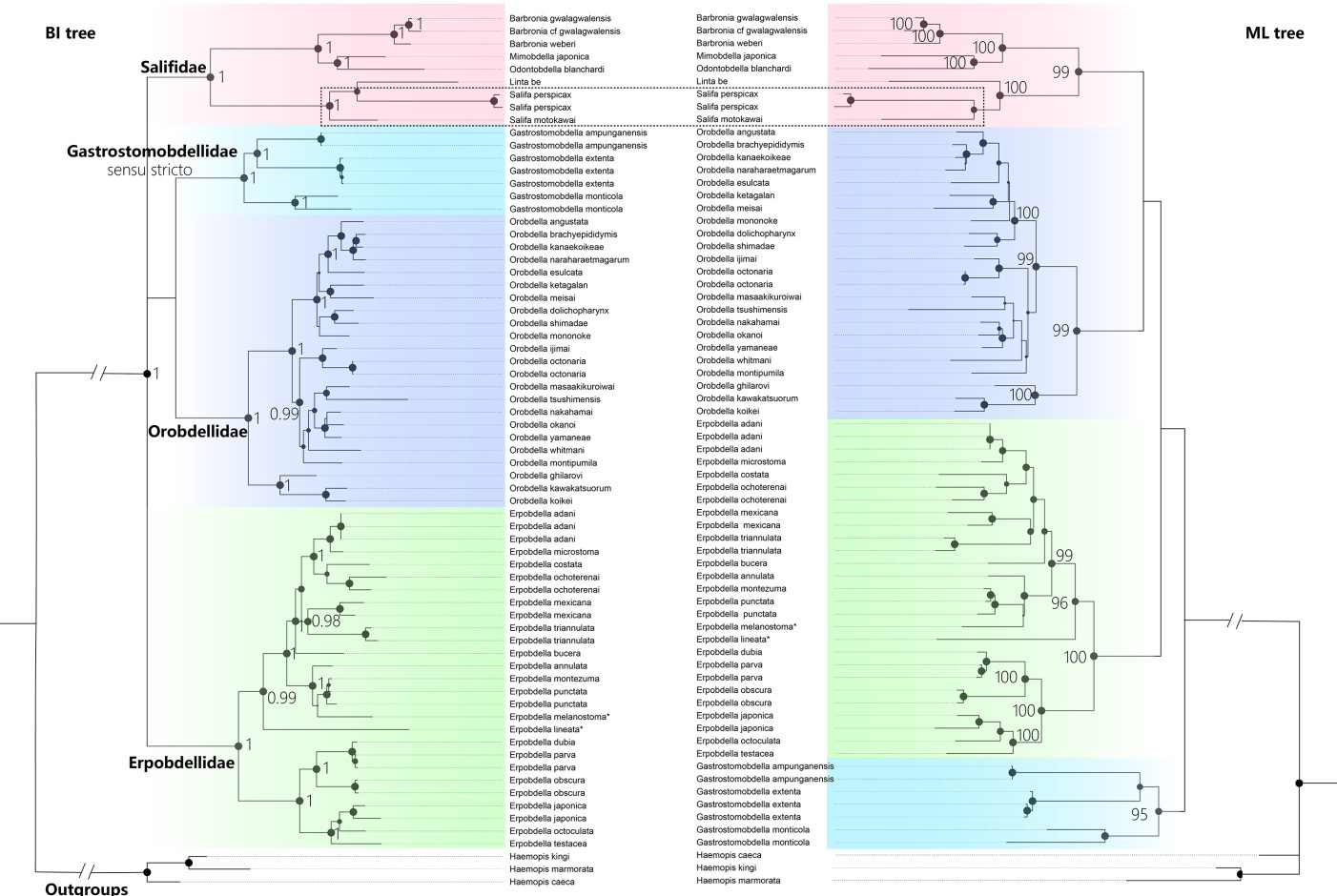

**Figure 5** **The Comparison of BI and ML trees constructed using a concatenated data of both mitochondrial genes (COI, 12S rDNA) and nuclear markers (18S and 28S rDNA), respectively.** The monophyly of the genus *Barbronia* and four Erpobdelliformes families (Salifidae represented by five genus, Orobdellidae, Gastrostomobdellidae *sensu stricto* and Erpobdellidae) were well supported. Two *Erpobdella* species name with an asterisk (*) were the synonym of *Dina lineata* and *Mooreobdella melanostoma*, respectively. The size of circles indicates that either posterior probabilities or the bootstrap values of corresponding nodes were estimated in BI or ML analyses. The 28S rDNA of *B. gwalagwalensis* (AY786449) was not included in the current analyses, since it was a partial conserved region of 28S rDNA but significantly different from other 28S rDNA sequences of *Barbronia* listed in the Table 1.

is a well-supported monophyletic group. The monophyly of Erpobdelliformes was well supported in Bayesian Inference (BI) analysis, with a posterior probability (PP) of 1, but it was not corroborated by the maximum likelihood (ML) analysis.

## DISCUSSION

To classify *Barbronia* cf. *gwalagwalensis* as a distinct species, additional evidence is needed since cryptic speciation is common among clitellates (*Erséus & Gustafsson, 2009*; *Liu et al., 2017*; *Martinsson & Erséus, 2021*). *Van Haaren et al. (2004)* identified one Dutch specimen as belonging to the *B. assiuti/weberi* complex, acknowledging that distinguishing the species or identifying it based solely on the morphological characteristics of this single specimen is also not feasible, which highlights the need for careful distinction between individual sample identities and broader species classification. The evidence for new

species should encompass both multiple loci-based genetic approaches and morphological analyses from numerous specimens (*Dellicour & Flot, 2015*; *Jorger & Schrodl, 2013*; *Stengel et al., 2022*; *Sukumaran, Holder & Knowles, 2021*). Initially, this specimen was morphologically identified as the common species *B. weberi* due to the observation of accessory gonopores (the anterior and posterior ones separately close to the male and female gonophores). The presence of two accessory gonopores in this specimen rules out the possibility of it being *B. zhejiangica* or *B. yunnanensis* (*Yang, Wang & Zhang, 1997*). Molecular results show that this specimen is more closely related to *B. gwalagwalensis* and the Korean *Barbronia* specimen (KF966549), the latter of which has been tentatively assigned to *Barbronia* cf. *zhejiangica* by *Klass et al. (2021)*. Interestingly, accessory gonopores have also been observed in *B. gwalagwalensis* (*Westergren & Siddall, 2004*). However, there are no previous records of *B. gwalagwalensis* in China. Unfortunately, a thorough morphological examination of this small specimen is unfeasible due to the complete utilization of its body for DNA extraction, a necessary step for constructing a high-quality library in next-generation sequencing. Therefore, the current classification for this specimen is as the taxon *B.* cf. *gwalagwalensis*, rather than describing it as a new species. This situation highlights the potential for *B. gwalagwalensis* to have a broader distribution than previously thought. Initially identified in South Africa, this species has also been found in Myanmar, as noted by *Klass et al. (2021)*. Further studies, involving more samples from across Asia and Europe and detailed morphological analysis, are required to draw definitive conclusions.

Over last two decades, species identification based on molecular data has indeed provided a valuable complement to morphological taxonomy (*Mahadani et al., 2022*), aided by the increasing availability of genetic techniques (*Dellicour & Flot, 2015*). In the current study, seven specimens of *B. weberi* constituted a well-supported clade in the COI phylogenetic results, this clade is also recognized as one valid species both in bGMYC and bPTP analyses (Fig. 3). The current species delimitation results support that the previously proposed taxa (*B. arcana*, *B. weberi* formosana and *B. wuttkei*) are synonyms of *B. weberi* rather than valid species, which is generally consistent with previous studies (*Klass et al., 2021*; *Oceguera-Figueroa et al., 2011*). The COI sequence similarity among the seven individuals within the *B. weberi* group (clade A in Fig. 3), collected from South Africa, the United States, Mexico, Costa Rica, and Germany are nearly identical. The lack of molecular differentiation (small p-distance) between them can be explained by the large effective population size but low rate of mutation, caused by both the relatively large distribution of invasive *B. weberi* and the potentially low speciation rate (*i.e.*, reproduce cocoon without cross-fertilization as mentioned in the introduction). However, it is important to note that a gene tree is not always consistent with a species tree when population sizes are large and speciation rates are high (*Dellicour & Flot, 2015*), single locus-based methods may not be sufficient for delimiting specimens listed in the group of *B. gwalagwalensis*, especially considering the relatively high genetic diversity between specimens sampled from southern Africa, France, Korea, and China (Fig. 4). On the basis of all available *Barbronia* data listed in Table 1, all *Barbronia* specimens tend to lump into one taxon by applying multiple loci-based species delimitation method. This is likely due to insufficient data. In this

scenario, utilizing data from both the mitochondrial genome and nuclear loci collected from each specific *B. gwalagwalensis* specimen (listed in Table 1) is expected to provide a more promising solution for resolving the species delimitation issue in *B. gwalagwalensis*, rather than relying solely on the partial mitochondrial gene COI.

Mitochondrial genomes and full-length nuclear rDNA have been now routinely applied to assess species boundaries and deep relationships in many phylogenetic studies due to obtaining sufficient data efficiently through NGS (*Jia et al., 2023*; *Moreno-Carmona et al., 2023*; *Prada et al., 2023*). However, a complete mitochondrial genome of any *Barbronia* species or of the family Salifidae remains unpublished, highlighting a significant gap in our understanding of their systematics. This lack of such data presents a challenge in accurately determining species delimitation and phylogenetic relationships within these groups, underscoring the need for more comprehensive genomic research. In the current study, the first complete mitochondrial genome of salifid leeches is assembled, and compared with whole or incomplete mitochondrial genome sequences of erpobdellids (Fig. 1 and Table 3), representing three nominal species and one unrecognized species in the GenBank. The specimen named *Erpobdella octoculata* (KC688270 and NC_023927) is likely to be a misidentified specimen or the outcome of sequencing contamination (*Oceguera-Figueroa et al., 2016*). Clearly, it is far from insufficient using these data to estimate either the species delimitation of *Barbronia* species or the phylogenetic relationships of *Barbronia* within Erpobdelliformes (Arhynchobdellida, Hirudinea). Therefore, four conventional molecular markers used in many previous studies of Erpobdelliformes are collected and analyzed, including full-length 18S and 28S rDNA sequences and COI and 12S rDNA sequences extracted from the newly acquired mitochondrial genome. The four well-supported clades, *i.e.*, Salifidae, Orobdellidae, Gastrostomobdellidae *sensu stricto*, and *Erpobdella*, are clearly distinguished in the current molecular phylogenetic results. The *Barbronia* clade is sister to a clade consisting of *Odontobdella blanchardi* and *Mimobdella japonica* with well supported in the current study (Fig. 5), and the monophyly of the *Barbronia* is congruent with previous research (*Anderson, Braoudakis & Kvist, 2020*; *Klass et al., 2021*; *Nakano et al., 2018*; *Nakano & Nguyen, 2015*; *Oceguera-Figueroa et al., 2011*). However, the *Salifa* genus represented by two species is not monophyletic in the current study and in *Nakano et al. (2018)* result, which is not consistent with previous studies (*Klass et al., 2021*; *Nakano & Nguyen, 2015*). The Orobdellidae clade is well supported, which is congruent with previous studies (*Nakano, 2016a*, *2016b*, *2022*; *Nakano et al., 2018*; *Nakano & Nguyen, 2015*; *Nakano, Ramlah & Hikida, 2012*). Additionally, previous studies have shown that Erpobdella forms a well-supported clade within the Erpobdellidae family. However, incorporating more species from Gastrostomobdella, rather than just one, provides stronger support for a closer relationship between the Erpobdella clade and the Gastrostomobdellidae family (*Nakano et al., 2018*; *Nakano & Nguyen, 2015*). Despite these insights, the robust phylogenetic positions of four families within Erpobdellidae remains uncertain in the current study, these findings underscore the intricate evolutionary relationships within the Erpobdellidae family, indicating the need for further research using more data to fully elucidate the phylogeny of these leech genera. To solve these problems, further studies involving more sampled specimens, or at least simultaneously

obtaining more molecular data (*i.e.*, mitochondrial genomes and rDNAs) from a few valuable specimens using NGS are urgently needed. The forthcoming systematics study of *B. gwalagwalensis* and related species is expected to benefit significantly from the methods employed in our current research, particularly in acquiring mitochondrial genomes and full-length rDNA sequence. This approach effectively bridges gaps between sequences amplified with different primers, notably improving results in cases where conventional primers are less effective (*Siddall, 2002*; *Trontelj, Sket & Steinbrück, 1999*).

## CONCLUSIONS

The first complete mitochondrial genome, full-length 18S and 28S rDNAs of salifid leeches within the family Salifidae are provided. The species delimitation results based on COI data supported the view that the previously proposed taxa (*Barbronia arcana*, *B. weberi* formosana, and *B. wuttkei*) are synonyms of *B. weberi*. However, the taxonomic status of *B. gwalagwalensis* and *B.* cf. *gwalagwalensis* need to be further studied, necessitating extensive fieldwork across Asia and Europe. *Barbronia* formed a well-supported clade, including *B. weberi*, *B. gwalagwalensis*, and *B.* cf. *gwalagwalensis*, and *Barbronia* is sister to a well-supported clade comprising *Odontobdella* and *Mimobdella*. They constitute a strongly supported monophyletic group and are incorporated into the Salifidae family. Additionally, the families Salifidae, Gastrostomobdellidae *sensu stricto* (containing only the genus *Orobdella*) and Orobdellidae are all well-supported monophyletic clades. According to the results of this study, the strategy of obtaining both whole mitochondria and nuclear markers from extensively sampled salifid species using NGS is expected to elucidate the species diversity of *B. gwalagwalensis* and the evolutionary relationship of Salifidae.

## ACKNOWLEDGEMENTS

We are grateful to Alejandro Oceguera-Figueroa and the two anonymous reviewers for their invaluable feedback, which greatly contributed to both the refinement of this article and the incorporation of additional literature to bolster its content.

### Funding

This work was supported by grants from the Natural Science Foundation of China (No. 31801957 and No. 32100334), the Natural Science Foundation of Jiangsu Province (No. BK20181471 and No. BK20210897), the Natural Science Foundation of the Higher Education Institutions of Jiangsu Province, China (No. 18KJD180007) and the Science and Technology Project of Xuzhou (No. KC19015). The funders had no role in study design, data collection and analysis, decision to publish, or preparation of the manuscript.

### Grant Disclosures

The following grant information was disclosed by the authors:
Natural Science Foundation of China: 31801957 and 32100334.

Natural Science Foundation of Jiangsu Province: BK20181471 and BK20210897.
Natural Science Foundation of the Higher Education Institutions of Jiangsu Province,
China: 18KJD180007.
Science and Technology Project of Xuzhou: KC19015.

## Competing Interests

The authors declare that they have no competing interests.

## Author Contributions

- Yingkui Liu conceived and designed the experiments, analyzed the data, authored or
  reviewed drafts of the article, and approved the final draft.
- Xinxin Fu performed the experiments, prepared figures and/or tables, and approved the
  final draft.
- Yu Wang performed the experiments, prepared figures and/or tables, and approved the
  final draft.
- Jing Liu analyzed the data, authored or reviewed drafts of the article, and approved the
  final draft.
- Yong Liu analyzed the data, authored or reviewed drafts of the article, and approved the
  final draft.
- Chong Li conceived and designed the experiments, authored or reviewed drafts of the
  article, and approved the final draft.
- Jiajia Dong conceived and designed the experiments, prepared figures and/or tables,
  authored or reviewed drafts of the article, and approved the final draft.

## Data Availability

The data is available at NCBI: OQ339201, OQ269483, OQ269482.

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
