# Peer review of "Exploring Barbronia species diversity and phylogenetic relationship within Suborder Erpobdelliformes (Clitellata: Annelida)"

_PeerJ, doi:10.7717/peerj.17480_

## Round 0.1 · original submission · Major Revisions

Dear authors,

After obtaining the reviews of three referees, all agree that it is an interesting work because molecular information is being obtained from a species or species that have been little studied, however, all agree that there are several points to improve before the manuscript can be accepted, mainly they highlight errors in the English writing, for this reason, I suggest that a fluent speaker review it, as well as some suggestions to improve the analysis. For this reason, it is necessary to carry out Major Revisions.

Sincerely,

Armando Sunny.

Reviewer 1 ·

Basic reporting

The English is also a problem. I had a hard time understanding certain portions, and in general there are a lot of issues with the English even when the message is clear. I strongly suggest having an English editing service or friend help with this. I have pointed to some examples in my specific comments, but many more are present.

Experimental design

There are no images of the specimen used (I believe the whole specimen was used for sequencing), and little description of the morphology was given outside of mention of the accessory pore.

The large focus on species delimitation seems misguided and (in my opinion) should be removed. There are few specimens, they are geographically disparate, have limited data, and weren’t sampled for this type of work.

The mitochondrial genome is the largest contribution.

Validity of the findings

I would delete the species delimitation, or at least make it a small part of the paper.

Additional comments

Liu et al. present some new data and a reanalysis of GenBank data for Barbronia leeches. The main novelty in the paper seems to stem from sequence data related to a specimen labeled B. cf. gwalagwalensis. For this specimen, a mitogenome, standard Sanger data, and full length ribosomal markers were sequenced. These are good data to have added for working with these leeches. However, the paper has issues.

The title shouldn’t be “species diversity of B weberi“ because you can’t have species diversity within a species. The title in general should reflect the main efforts of the work, which to me is more the mitochondrial and other data addition. Mentioning exploring diversity within Barbronia would be fine too.

For the phylogenetic comparisons, maybe consider the following two papers, as they include relationships for some of the focal taxa:
https://doi.org/10.1206/3895.1
https://doi.org/10.1016/j.ympev.2019.106688

Here are some specific edits:

1. L25 change to “molecular markers”
2. L35 English writing needs work.
3. L57 “Barbronia is”
4. L70 English writing needs work.
5. L78 unclear what you mean.
6. L90 first instead of firstly.
7. L91 did this analysis include Salifidae leeches? If not, it isn’t surprising/especially relevant that Barbronia came out as related to Erpobdella (Dina is typically synonymized with Erpobdella, as you mentioned below).
8. L126 how was the specimen identified?
9. L201 and L203 delete “were”.
10. L212 double periods.
11. L213 change was to were.
12. L289 are you talking about taking data from one specimen and adding it to other sequences from different specimens? This would be very bad, given the inconsistency of the COI data.
13. L292 this point seems like one of the most important parts of the discussion, but I don’t know what you mean. Specifically, I don’t know what you are saying at the start of the paragraph. The English needs reworking for clarity. It becomes clearer late on in the paragraph, but then is still unclear about which taxa have accessory pores.
14. L327 “species delimitation”.
15. L370 delete this section if not using.

Reviewer 2 ·

Basic reporting

Dear Authors and Editor,

I have reviewed the manuscript entitled "Exploring the species diversity of B. weberi and the
phylogenetic relationship of Barbronia within
Suborder Erpobdelliformes (Clitellata: Annelida)" for PeerJ. I detail below my thoughts about the manuscript. But in general, I think the data that the authors add is very relevant to the scientific community, although I have a few concerns.

Major comments:
1. The authors sequenced the mitochondrial genome of one single specimen of Barbronia and all subsequent analyses were performed using only COI, in the case of species delimitations methods, and additional 12S, 18S, and 28S rDNA. The mitochondrial genome itself is an extremely valuable addition to public databases and it is certainly worth publishing, but as the authors mention there is a large lack of comparable mitochondrial genomes within Erpobdelliformes. This lack of comparative data makes their data even more valuable but limits their capacity to use their mitochondrial genome efficiently. In my opinion, the authors should make it evident that the species delimitation is not based on the whole mitochondrial genome, and the same analyses could be performed by applying Sanger sequencing. In some occurrences, the authors make it sound like the mitochondrial genome is essential to achieve the same results.

2. It should be much clearer which sequence is the new sequence and which ones are deposited in the published databases. This is extremely important because it is not clear how much new data the authors provide and how the addition of this data changes our understanding of leech evolution.

3. Since the authors are proposing the publication of a mitochondrial genome, some sort of comparison between mitochondrial genomes within leeches should be provided, otherwise this data becomes lost in the manuscript.

4. The species delimitation analyses are performed using only a few individuals of each species. Are there any problems related to this very small dataset?

5. I am afraid that Erpobdella is not a good root for the phylogeny shown in figure 3. In Nakano et al. (2012), Erpobdellidae falls within Erpobdelliformes, but their exact phylogenetic position can be defined. In this analysis, using Erpobdella as a root, is forcing the monophyly of Gastrostomobdellidae + Orobdellidae + Salifidae. Therefore, the authors do not evaluate the monophyly of these three families as they claim. A proper root should be used!

Other comments:
1. In many instances it is not clear what the authors are trying to communicate. I recommend a major revision of the manuscript to correct some grammar and style.

2. Species names are not correctly written in many instances. For example, the "Barbronia sp" and "B. cf gwalagwalensis" without the '.' after 'sp' and 'cf' occurs several times.


For these reasons, I recommend at major revision to the manuscript.

Experimental design

Dear authors,

As mentioned above, I am not sure how the very small dataset used for the species delimitation analyses can bias the results.

Moreover, the phylogeny in Figure 3 is not properly rooted and the monophyly of Gastrostomobdellidae + Orobdellidae + Salifidae is an artifact of the lack of a proper root.

Validity of the findings

Dear authors,

My major concern about this manuscript is the lack of a proper root in Figure 3, which invalidates the claim that Gastrostomobdellidae + Orobdellidae + Salifidae is monophyletic.

·

Basic reporting

This manuscript requieres a review to improve the English.
The references are fine, just some minor problems regarding some family names of the authors.
The structure is in general fine, however my major concern is that in the methods section is not clear how many leech specimens were collected (I think is only one!), it is not clear how the specimen was fixed/preserved, identified.
Raw data is shared.

Experimental design

The authors generated new data of a leech member of a very poorly known group. In this sense, the manuscript is important given the generation of a complete mitochondrial genome of an unidentified species of Barbronia. On the other hand, the authors provided some "sophisticated" method to delimit species based on molecular data. In general, these methods are interesting and their conclusions agree with comments provided by previous authors.
I strongly recommend that a section of the morphological traits of the leeches be included in the results. Authors need to proof that they are working with the leech species that they named.

Validity of the findings

In general, it is important to generate new molecular data representing poorly studied groups, and this is probably the most important aspect of this manuscript (generation of one mitochondrial genome and two nuclear markers). However, the results are not very interesting, especially because there is nothing new in the mitogenome and their "taxonomic" conclusions just confirm previous studies.

Additional comments

The morphological description of the leech specimen is very poor. It is hard to understand why the authors only collected one specimen, and it is hard to understand why the compete specimen was used for molecular techniques. Without voucher specimens it is hard to validate the results and it is impossible to validate the species identity. I strongly recommend that the authors collect more leeches and check their morphological traits, not only molecular data with sophisticated methods. It is important to provide a more detailed description of the collection locality and pictures/drawings of the leeches. Also, it is important to keep vouches specimens in scientific collections.

---

## Round 0.2 · Major Revisions

Dear authors,

Both reviewers comment that the phylogenetic tree must be rooted outside of Erpobdelliformes. E.g. a few Hirudiformes; see Tessler et al. 2018, please make this correction, so that your study can be publishable. therefore major corrections are required.

Best regards,

Armando Sunny

Reviewer 1 ·

Basic reporting

This has improved since my first round of review.

Experimental design

I still think the species delimitation work doesn't fully make sense. Another reviewer pointed this out in their review, highlighting that this type of work has difficulties with the small sample size used here. However, the authors do make some changes to lessen the problem. Ultimately, I leave this up to the editor.

The authors also seemingly did not understand the other reviewer's point about rooting the phylogenetic tree. The reviewer was suggesting a root outside of Erpobdelliformes. E.g. a few Hirudiformes; see Tessler et al. 2018 for an overview phylogeny of leeches. Also see Nakano papers that detail the relationships within these specific groups to compare.

Validity of the findings

This seems similar to the last review. Again, I personally am not sure about the species delimitation or the rooting choice.

Reviewer 2 ·

Basic reporting

Dear authors,

After rereviewing the manuscript entitled ‘exploring Barbronia species diversity and phylogenetic relationships within Suborder Erpobdelliformes (Clitellata: Annelida)’, I am left with no choice but to reject this manuscript. The main reason for this recommendations is the lack of a proper root for the phylogenetic analysis presented in Figure 5. The outgroup of a phylogenetic analysis cannot be an ingroup. This was already pointed out in the previous round of reviews and the authors show no good explanation for not including other lineages as outgroups.

I detailed my comments below, if that is of interest to you.


General comment 1. Grammar needs some attention, but it does not impact the general understanding of the manuscript.

General comment 2. subtitles are not standardized. For example, Some are written with the first letter of each word capitalized and others only the first letter of the subtitle is capitalized.

line 61. I think this is the first time that these species are mentioned. Authors should be included when they are mentioned for the first time as the ICZN states.

line 75. Which parasitic trematodes? You should be specific.

line 85. the ‘a’ of ‘B. arcana’ is not italicized.

line 86. Include a ‘.’ in the ‘sp’

line 76-88. is this paragraph really necessary?

line 122. I think the authors mean ‘assembled’

line 140. What were the parameters used in trommomatic?

line 143. parameters for MITOS2

line 141. parameters for GetOrganelle

line 147-150. I think this was already mentioned in the introduction and I dont think this should be stated in the methods. Instead, the authors should provide a table listing all sequences used in their study, their respective collection site and source.


lines 146–176. ‘Single-locus based species delimitation’. The authors should provide detailed information on the parameter values used for each of the analyses performed. For example, there is no information about the alignment of matrices used.
Data availability. It is desirable that the authors provide matrices to be checked and that this information is made publicly available.
Please state which terminals were used as outgroups.


line 175-177. How would the software work with species that are only distantly related such as M. japonica and O. blanchardi?

line 184-185. ‘Phylogenetic analysis using multiple loci’. parameters used for alignment of matrices should be provided. State which terminals were used as outgroup.

line 188-189. ‘Phylogenetic analysis using multiple loci’. All parameters used in IQ-TREE and MrBayes should be provided. Moreover, the best fitting model should be stated.


lines 203-217. ‘General Features of Mitochondrial Genomes and nuclear rDNAs’. I think these results are a bit apart from the whole manuscript.


‘Phylogenetic analysis using multiple loci’. Figure 5. How were these trees rooted? I dont think these trees should be rooted using the Erpobdella clade. The outgroup should not belong to the ingroup. This is evidenced by the taxonomic confusion between species belonging to Barbronia and Erpobdella (as stated by the authors) and the close relationship between these two genera. Especially if the authors are interested in the relationships of Erpobdelliformes as they state in their introduction. A proper outgroup should be used in the phylogenetic analyses.
The authors even state ‘‘In addition, the basal phylogenetic placement of Erpobdella within the Erpobdelliformes may be controversial when different outgroups are used (Nakano et al. 2018; Nakano & Nguyen 2015)”.

Experimental design

no comment

Validity of the findings

All claims regarding the relationship within Erpobdelliformes are not valid because the authors did not use a proper outgroup to root their trees.

Additional comments

no comment

---

## Round 0.3 · accepted · Accept

Dear Authors,

Following the last review, both reviewers expressed concerns regarding the lack of rooting in the phylogenetic tree, which raised doubts about the validity of the results. However, upon reviewing the revised manuscript, I am pleased to see that you have addressed this issue effectively. As such, I am confident in recommending acceptance of the manuscript for publication.

Thank you for carefully considering the previous observations and for submitting your manuscript to PeerJ.

Best regards,

Armando Sunny